# Effect of Microbial Inoculation on Carbon Preservation during Goat Manure Aerobic Composting

**DOI:** 10.3390/molecules26154441

**Published:** 2021-07-23

**Authors:** Jiawei Lu, Jingang Wang, Qin Gao, Dongxu Li, Zili Chen, Zongyou Wei, Yanli Zhang, Feng Wang

**Affiliations:** 1Institute of Goats and Sheep Science, Nanjing Agricultural University, NO. 1 Weigang, Nanjing 210095, China; 2018105025@stu.njau.edu.cn (J.L.); T2017154@njau.edu.cn (Q.G.); 2019105030@stu.njau.edu.cn (D.L.); 2020805121@stu.njau.edu.cn (Z.C.); 2Research Centre of Haimen Oats, Nanjing Agricultural University, Haimen 216121, China; 2018105027@stu.njau.edu.cn; 3Taicang Agricultural and Rural Science & Technology Service Center, Taicang 215400, China; 2018105026@stu.njau.edu.cn

**Keywords:** carbon preservation, composting, bacterial community, *Bacillus subtilis*, *Bacillus licheniformis*

## Abstract

Carbon is the crucial source of energy during aerobic composting. There are few studies that explore carbon preservation by inoculation with microbial agents during goat manure composting. Hence, this study inoculated three proportions of microbial agents to investigate the preservation of carbon during goat manure composting. The microbial inoculums were composed of *Bacillus subtilis*, *Bacillus licheniformis*, *Trichoderma viride*, *Aspergillus niger*, and yeast, and the proportions were B1 treatment (1:1:1:1:2), B2 treatment (2:2:1:1:2), and B3 treatment (3:3:1:1:2). The results showed that the contents of total organic carbon were enriched by 12.21%, 4.87%, and 1.90% in B1 treatment, B2 treatment, and B3 treatment, respectively. The total organic carbon contents of B1 treatment, B2 treatment, and B3 treatment were 402.00 ± 2.65, 366.33 ± 1.53, and 378.33 ± 2.08 g/kg, respectively. B1 treatment significantly increased the content of total organic carbon compared with the other two treatments (*p* < 0.05). Moreover, the ratio of 1:1:1:1:2 significantly reduced the moisture content, pH value, EC value, hemicellulose, and lignin contents (*p* < 0.05), and significantly increased the GI value and the content of humic acid carbon (*p* < 0.05). Consequently, the preservation of carbon might be a result not only of the enrichment of the humic acid carbon and the decomposition of hemicellulose and lignin, but also the increased OTU amount and *Lactobacillus* abundance. This result provided a ratio of microbial agents to preserve the carbon during goat manure aerobic composting.

## 1. Introduction

The goat industry has developed quickly in China, thus a large amount of goat manure is produced. Goat manure is considered a very valuable resource because of the abundant nutrients contained in it and the ability of using it to improve soil fertility [1,2].

Composting feeds nutrients and organic matter back into soils to improve crop growth and can decompose organic matter into humic substances [3,4]. The process of composting is affected by both environmental and biological factors, such as temperature, humidity, oxygen content, organic matter composition, and the presence of microorganisms [5]. Aerobic composting reduces the negative environmental consequences of biological waste and efficiently transforms biological waste into valuable fertilizers [6]. Nevertheless, CO_2_ and CH_4_ would be released along with the composting process, which results in the reduction of carbon in compost and an adverse effect on the environment [7]. Some previous studies stated that composting led to a loss of 14–59% of total organic carbon compared with the initial TOC content [8,9,10].

Various studies have found that the way to reduce the loss of carbon and nitrogen during the composting process is to use physical absorbents and some chemicals to absorb CH_4_ and NH_3_ [11,12,13]. Inoculating exogenous microbial agents has an important role in preserving carbon during the composting period. *Bacillus subtilis* at 0.2% could inhibit the mineralization, thereby enriching the contents of total organic carbon and humic substances. pH and microorganisms can also affect the preservation of carbon during composting [14]. Furthermore, *Bacillus subtilis* could degrade the cellulose and decrease the pH during the composting process [15,16].

*Bacillus licheniformis* BT575 exhibited great potential as an inoculant for compost bioaugmentation [17]. *Trichoderma viride* could produce cellulase to decompose cellulose materials [18]. It has been reported that composts with the presence of *Aspergillus niger* can be either applied to soil as an organic soil amendment or used as an organic substrate for seedlings without the risk of pathogens spreading in the environment and damaging crops [19].

At present, there are relatively few studies on the loss of carbon with microbial inoculation of goat manure compost. The mechanism of reducing the carbon loss of different microbial inoculation on the process of goat manure composting remains unknown. In this study, the effects of microbial inoculation on the carbon content and bacterial community of goat manure compost during different periods of aerobic composting were investigated. This study selected the proportion of microbial agents that could reduce the loss of carbon and explored the mechanism of preserving carbon during composting. The results reveal a new view on the preservation of carbon during goat manure composting. This provides a scientific theoretical method for the production of goat manure organic fertilizer, so as to more efficiently and harmlessly deal with goat manure and agricultural waste.

## 2. Materials and Methods

### 2.1. Materials and Composting Experimental Design

The compost materials used in this experiment were goat manure and rape straw, collected from Jiangsu Jinsheng Goat Breeding Technology Development Company. The goat manure was fresh and rape straw was dried and stored in warehouse before composting, then the rape straw was cut into 2–5 cm pieces whose physicochemical properties were shown in Table 1. Microbial inoculants were provided by Kangyuan Biotechnology Company. The *Bacillus subtilis* and *Bacillus licheniformis* were present at 100 billion colony-forming units per gram, and *Trichoderma viride*, *Aspergillus niger*, and yeast were present at 10 billion colony-forming units per gram. *B. subtilis* was dissolved in 0.5 mL of Luria–Bertani culture (LB) broth, then inoculated on an LB agar medium, which was then cultured in an inoculator at 37 °C for 24 h. The method of preparation was the same as that of *B. licheniformis*. *T. viride* was dissolved in 0.5 mL of potato broth and inoculated on a potato agar medium, then incubated at 30 °C for 24 h. The preparation method of *A. niger* was the same as that of *T. viride*. After this, yeast was dissolved in 0.5 mL of wort broth, inoculated on a wort agar medium, and subsequently cultured at 25 °C for 24 h. These bacteria were used as inoculants in compost. According to the characteristics and the optimum temperature of each inoculation, the total proportions of the microbial inoculants were 1:1:1:1:2 (B1), 2:2:1:1:2 (B2), and 3:3:1:1:2 (B3).

The mixed materials were transported to the fermentation area with goat manure as the main material and rape straw as the secondary material and then adjusted to a moisture content of approximately 55–65% and C:N ratio of 20:1 [20]. Thus, composting reactors were built whose length, width, and height were 1.5 m × 1.0 m × 1.5 m, respectively, and each pile was approximately 1 ton. The number of microbial agents in each composting treatment was 3% [20]. Plastic coverings were used to not only conserve heat but also maintain the necessary temperature in winter, and a turning machine was used to turn each pile to ensure that the compost materials were evenly mixed. Turning by rotary stirring was performed every three days during the first 10 days and every 10 days for the next 50 days. The reactors were turned over on days 1, 4, 8, 11, 21, 29, 39, 49, and 60 to encourage aeration and increase the oxygen content. Measurements were performed at the time of turning of each pile [21].

### 2.2. Sample Collection and Analysis of Physicochemical Indicators

A series of 600 g samples were collected from the upper, middle, and lower sections of the composting pile after each turn, then mixed homogeneously [22]. These samples were divided into two parts weighing 0.3 kg each. One part was used for measurements of physicochemical indicators and was preserved at 4 °C, and the other part was used for examination of the bacterial community and was stored at −80 °C. Samples were collected on days 1, 4, 8, 11, 21, 29, 39, 49, and 60. The period of composting experiment was approximately 60 days.

The temperature was measured twice a day (9:30 and 15:30) in the upper layer (10–20 cm), middle layer (50–70 cm), and lower layer (100–120 cm) of the compost piles, and the ambient temperature was simultaneously measured. The measurement was repeated three times using a thermometer (Shanghai, China), and the average value of the three layers was taken as the average temperature that day. The moisture content was detected by drying at 105 °C to a constant weight [23]. Compost samples and deionized water were mixed at 1:10 (*m*/*v*) (25 °C), shaken in a shaker at a speed of 200 r/min for 2 h, and allowed to stand for 20 min, after which the pH and EC were detected. Seven milliliters of supernatant were placed into a Petri dish lined with 2 sheets of filter paper. The control group received deionized water, and 20 full Chinese cabbage seeds were evenly placed on the filter paper then cultivated in an incubator for 48 h (30 °C). The germination rate and root length of Chinese cabbage seeds were measured. The germination index (GI) was calculated according to the formula [24]:(1)$$\mathrm{GI}=\frac{\mathrm{Seed}\mathrm{Germination}\times \mathrm{Root}\mathrm{Length}\mathrm{in}\mathrm{Treatment}}{\mathrm{Seed}\mathrm{Germination}\times \mathrm{Root}\mathrm{Length}\mathrm{in}\mathrm{Control}}\times 100\%$$


Total nitrogen content was detected with Kjeldahl nitrogen analyzer and the total organic carbon was determined by total organic carbon analyzer. These measurements were compared to get the C/N ratio. The contents of ammonium nitrogen and nitrate nitrogen were measured by a flow injection analysis system [25]. The contents of humic substance carbon, humic acid carbon, and fulvic acid carbon were analyzed by SHIMADZU TOC-Vcph analyzer [26]. The contents of cellulose, hemicellulose, and lignin were determined by detecting the natural detergent fiber (NDF), acid detergent fiber (ADF), and acid detergent lignin (ADL), and a fiber analyzer (ANKOM A200i) was used in accordance with the means of [27]. All the above indicators were detected on a dry basis.

### 2.3. Bacterial Composition Analysis

#### 2.3.1. DNA Extractions

DNA from different samples was extracted using the E.Z.N.A.^®^ Stool DNA Kit (D4015, Omega, Inc., San Antonio, TX, USA) according to the manufacturer’s instructions. Nuclear-free water was used as a blank. The total DNA was eluted in 50 μL of Elution buffer and stored at −20 °C until measurement in the PCR by LC-Bio Technology (Hangzhou, China).

#### 2.3.2. PCR Amplification and 16S rDNA Sequencing

The hypervariable V3–V4 region of the prokaryotic (bacterial and archaeal) small-subunit (16S) rRNA gene was amplified with primers 341F (5′-CCTACGGGNGGCWGCAG-3′) and 805R (5′-GACTACHVGGGTATCTAATCC-3′). The 5′ ends of the primers were tagged with specific barcodes per sample and sequenced using universal primers. PCR amplification was performed on a total volume of 25 μL reaction mixture containing 25 ng of template DNA, 12.5 μL PCR Premix, 2.5 μL of each primer, and PCR-grade water to adjust the volume. The PCR conditions to amplify the prokaryotic 16S fragments consisted of an initial denaturation at 98 °C for 30 s; 32 cycles of denaturation at 98 °C for 10 s, annealing at 54 °C for 30 s, and extension at 72 °C for 45 s; and then a final extension at 72 °C for 10 min. The PCR products were confirmed with 2% agarose gel electrophoresis. Throughout the DNA extraction process, ultrapure water, instead of a sample solution, was used to exclude the possibility of false-positive PCR results as a negative control. The PCR products were purified by AMPure XT beads (Beckman Coulter Genomics, Danvers, MA, USA) and quantified by Qubit (Invitrogen, Waltham, MA, USA). The amplicon pools were prepared for sequencing and the size and quantity of the amplicon library were assessed on Agilent 2100 Bioanalyzer (Agilent, Santa Clara, CA, USA) and with the Library Quantification Kit for Illumina (Kapa Biosciences, Woburn, MA, USA), respectively. The libraries were sequenced on NovaSeq PE250 platform.

#### 2.3.3. Processing of Sequencing Data

Samples were sequenced on an Illumina NovaSeq platform according to the manufacturer’s recommendations, provided by LC-Bio. Paired-end reads were assigned to samples based on their unique barcode and truncated by cutting off the barcode and primer sequence. Paired-end reads were merged using FLASH. Quality filtering on the raw tags was performed under specific filtering conditions to obtain high-quality clean tags according to the fqtrim (V 0.94). Chimeric sequences were filtered using Vsearch software (V 2.3.4). Sequences with ≥97% similarity were assigned to the same operational taxonomic units (OTUs) by Vsearch (V 2.3.4). Representative sequences were chosen for each OTU, and taxonomic data were then assigned to each representative sequence using the RDP (Ribosomal Database Project) classifier. OTUs abundance information was normalized using a standard of sequence numbers corresponding to the sample with the least sequences. Alpha diversity was applied in analyzing complexity of species diversity for a sample, including Chao1, Observed_species, Goods_coverage, Shannon, Simpson, and all these indices in our samples were calculated with QIIME (Version 1.8.0). Blast was used for sequence alignment, and the OTU representative sequences were annotated with RDP (ribosome database) and NCBI-16S database for each representative sequence. Other diagrams were implemented using the R package (V 3.4.4).

### 2.4. Statistical Analysis

Data were statistically analyzed with one-way ANOVA using SPSS 23.0 (*p* < 0.05). The results were shown as the Mean ± Standard Error (M ± SE). GraphPad Prism 8.0 was used for making graphs.

## 3. Results

### 3.1. Changes in Physicochemical Indicators

The trends of temperature were almost the same among the three treatments (Figure 1A). The temperature fluctuated after each turning. The maximum temperatures in B1 treatment, B2 treatment, and B3 treatment were 67.07 ± 1.75, 70.57 ± 1.53, and 68.37 ± 1.93 °C, respectively, whereas there were no significant differences among the three treatments (*p* > 0.05). The composting days on which the temperature exceeded 50 °C in B1 treatment, B2 treatment, and B3 treatment were 11 d, 15 d, and 10 d. The temperatures in the three treatments decreased to an ambient temperature on day 60. Microbial agents in the compost accelerated the reproduction of microorganisms, releasing a large amount of heat, thereby increasing the temperature in the pile.

The moisture contents in three treatments reflected a similar tendency to gradually decrease (Figure 1B). The initial moisture contents of three treatments were approximately 60%. The average moisture contents of B1 treatment, B2 treatment, and B3 treatment were 40.65 ± 13.14%, 42.64 ± 8.70%, and 42.33 ± 10.49%, respectively. When the composting ended, the moisture contents of B1 treatment, B2 treatment, and B3 treatment were 27.86 ± 0.20%, 33.40 ± 0.53%, and 30.27 ± 0.25%, respectively. Furthermore, the moisture content of B1 treatment was significantly lower than that of the other two treatments (*p* < 0.05). It may be that less addition of *Bacillus subtilis* and *Bacillus licheniformis* enlarged the decomposition rate of organic matter by microorganisms, as the water evaporated with heat in compost, causing a reduction in moisture content.

The trends of pH in the three treatments were almost the same (Figure 1C). The pH first increased and then decreased, finally reaching approximately 8.5. Across the entire composting period, the average pH values of B1 treatment, B2 treatment, and B3 treatment were 8.96 ± 0.23, 9.17 ± 0.32, and 9.08 ± 0.25, respectively. At the end of the composting process, the pH values of B1 treatment, B2 treatment, and B3 treatment were 8.50 ± 0.02, 8.64 ± 0.01, and 8.57 ± 0.02, respectively. Additionally, the pH value of B1 treatment was significantly lower than that of the other two treatments (*p* < 0.05).

Electrical conductivity revealed the toxicity of the compost, with an EC value less than 4.0 ms/cm contributing to the application of the compost [28]. The EC values of the three treatments changed dynamically (Figure. 1D). When the composting ended, the EC values of B1 treatment, B2 treatment, and B3 treatment were 3.22 ± 0.32, 4.93 ± 0.17, and 5.12 ± 0.25 ms/cm, respectively. The EC value of B1 treatment was significantly lower than that of the other two treatments (*p* < 0.05). It could be speculated that B1 treatment efficiently decomposed the toxic substances in the compost, leading to a reduction in the EC value.

The germination index revealed the maturity of compost via the effect of the compost extract on the growth of plant seeds [29]. Figure 1E showed that the GI value declined in B1 treatment on day 1; however, in both B2 and B3 treatments, the GI values began to decrease on day 8. The reactor produced toxic and harmful substances during the thermophilic stage, which was not beneficial to the growth of plants. Furthermore, the GI value of B1 treatment reduced seven days earlier than that of the other treatments, suggesting that excessive addition of *Bacillus subtilis* and *Bacillus licheniformis* would prolong the time in the thermophilic stage to kill toxic and harmful substances. When the composting ended, the GI values of B1 treatment, B2 treatment, and B3 treatment were 82.14 ± 0.03%, 80.10 ± 0.06%, and 80.39 ± 0.68%, respectively. The GI value of B1 treatment was significantly higher than that of B2 and B3 treatments after composting (*p* < 0.05). According to the results, microbial inoculation at the 1:1:1:1:2 ratio can not only strongly eliminate toxicity but also enhance the maturity and quality of compost, making it more suitable for compost application.

### 3.2. Changes in Nitrogen

The C/N ratios in the three treatments were almost the same, and all decreased after composting (Figure 2A). During the overall composting process, the C/N ratios of B1 treatment, B2 treatment, and B3 treatment were 13.83 ± 3.16, 13.82 ± 2.05, and 15.49 ± 1.58, respectively. The C/N ratios of the three treatments were not significantly different throughout the composting period (*p* > 0.05). When the composting ended, the C/N ratios of B1 treatment, B2 treatment, and B3 treatment were 14.97 ± 0.15, 14.03 ± 0.15, and 14.03 ± 0.06, respectively.

Figure 2B showed that the trends of total nitrogen in the three treatments were the same, with a small increase in the mesophilic stage and then a decrease before continuing to increase in the thermophilic stage and ultimately decreasing once more. The compost entered the mesophilic stage from the 4th day. The microorganisms vigorously consumed nitrogen into ammonia in the compost, resulting in the reduction of total nitrogen content during this phase. Subsequently, the compost entered the cooling phase from the 11th day, the ammonification was weakened leading to an increase in total nitrogen content. The total nitrogen contents of the three treatments were basically the same at the initial stage of composting, and it increased slightly after the completion of composting. Across the overall composting process, the total nitrogen contents of B1 treatment, B2 treatment, and B3 treatment were 24.34 ± 2.97, 25.52 ± 2.73, and 23.16 ± 2.99 g/kg, respectively. No significant differences in total nitrogen contents were detected between the three treatments during the composting process (*p* > 0.05). When the composting ended, the total nitrogen contents of B1 treatment, B2 treatment, and B3 treatment were 27.59 ± 0.03, 26.15 ± 0.02, and 26.10 ± 0.02 g/kg, respectively. The total nitrogen content of the B1 treatment was significantly higher than that of the other two treatments after composting (*p* < 0.05). Consequently, B1 could inhibit the volatilization of ammonia in compost.

Figure 2C shows that the ammonium nitrogen contents of all three treatments gradually decreased from the 21st day to the 60th day. The rise of ammonium nitrogen is because the nitrogenous organic compound was strongly degraded during the early stage. Whereafter, the ammonium nitrogen transformed into ammonia resulting in a decrease in ammonium nitrogen. During the whole composting period, the maximum values of ammonium nitrogen in B1 treatment, B2 treatment, and B3 treatment were 0.19 ± 0.003, 0.16 ± 0.002, and 0.16 ± 0.003 g/kg, respectively. The maximum value of ammonium nitrogen in B1 treatment was significantly higher than that in the other two treatments (*p* < 0.05). When the composting ended, the ammonium nitrogen contents of B1 treatment, B2 treatment, and B3 treatment were 0.03 ± 0.001, 0.04 ± 0.001, and 0.01 ± 0.001 g/kg, respectively. The ammonium nitrogen content of B3 treatment was significantly lower than that of the other two treatments after composting (*p* < 0.05).

Figure 2D showed that the nitrate nitrogen contents of the three treatments first increased and then decreased. The ammonia-oxidizing bacteria conducted nitrification and transformed the nitrogenous organic compound into nitrate nitrogen, leading to an increase in nitrate nitrogen. Subsequently, the decreased nitrification caused a reduction in nitrate nitrogen content. Across the entire composting period, the nitrate nitrogen contents of B1 treatment, B2 treatment, and B3 treatment were 0.17 ± 0.05, 0.16 ± 0.04, and 0.15 ± 0.05 g/kg, respectively. There were no significant differences in the nitrate nitrogen content of the three treatments during the whole composting period (*p* > 0.05). When the composting ended, the nitrate nitrogen contents of B1 treatment, B2 treatment, and B3 treatment were 0.21 ± 0.01, 0.13 ± 0.01, and 0.05 ± 0.01 g/kg, respectively. The content of nitrate nitrogen in B1 treatment was significantly higher than that in the other two treatments when the composting ended (*p* < 0.05).

### 3.3. Changes in Carbon

The variations in the total organic carbon in three treatments were almost the same, exhibiting first a decrease and then an increase (Figure 3A). The microorganisms multiplied utilizing organic carbon as a source of energy, which led to the reduction of the total organic carbon concentration. When the compost entered the maturity stage, the humification was improved, leading to an increase in the total organic carbon concentration. The average total organic carbon contents of B1 treatment, B2 treatment, and B3 treatment were 341.67 ± 52.10, 349.56 ± 34.72, and 361.11 ± 29.47 g/kg, respectively. Although there were no significant differences among the three treatments during the whole composting period (*p* > 0.05). After day 60 of composting, the total organic carbon contents of B1 treatment, B2 treatment, and B3 treatment were 402.00 ± 2.65, 366.33 ± 1.53, and 378.33 ± 2.08 g/kg, respectively, being enriched by 12.32%, 4.87%, and 1.90%, respectively. The total organic carbon content of B1 treatment was significantly higher than that of B2 and B3 treatments when the composting ended (*p* < 0.05). Apparently, B1 treatment played a central role in preserving carbon in compost.

The humic substance carbon content in three treatments first reduced and then increased (Figure 3B). The microorganisms degraded humic substance carbon into other substances, therefore, the content of humic substance carbon decreased. When the compost entered the maturity phase, a large amount of humic substance carbon was formed during this phase, resulting in the content of humic substance carbon increasing. The average contents of humic substance carbon in B1 treatment, B2 treatment, and B3 treatment were 293.76 ± 28.36, 282.69 ± 23.58, and 295.39 ± 26.00 g/kg, respectively. No significant differences were detected in the average contents of humic substance carbon between the three treatments during the whole composting phase (*p* > 0.05).

The contents of the humic acid carbon in three treatments demonstrated the same trends, which fluctuated and increased from the initial contents (Figure 3C). The increase in the humic acid carbon contents was due to the transformation of fulvic acid carbon during the final composting phase. When the composting ended, the contents of humic acid carbon in B1 treatment, B2 treatment, and B3 treatment were 34.17 ± 0.15, 30.03 ± 0.15, and 21.57 ± 0.15 g/kg, respectively, and the content of the humic acid carbon in B1 treatment was significantly higher than that of B2 and B3 treatments (*p* < 0.05). Consequently, the ratio of *Bacillus subtilis*, *Bacillus licheniformis*, *Trichoderma viride*, *Aspergillus niger*, and yeast at 1:1:1:1:2 improved the content of the humic acid carbon in compost.

The variations in the fulvic acid carbon between the three treatments were almost the same, showing fluctuations and reductions from the initial contents (Figure 3D). The reduction of the fulvic acid carbon may have resulted from the formation of humic acid carbon. With regard to the loss rate of the fulvic acid carbon, 16.05%, 58.23% and 34.53% of the initial fulvic acid carbon contents were decomposed in B1, B2, and B3 treatments, respectively. The average contents of fulvic acid carbon in B1 treatment, B2 treatment, and B3 treatment were 48.71 ± 10.95, 38.51 ± 12.27, and 49.10 ± 11.79 g/kg, respectively. There were no significant differences in the contents of fulvic acid carbon between the three treatments during the whole composting phase (*p* > 0.05).

### 3.4. Changes in Fiber

The contents of the cellulose first sharply reduced, then slowly rose, and gradually stabilized (Figure 4A). The cellulose was decomposed by hydrolase in compost during the composting process. At the end of composting, the contents of the cellulose in B1 treatment, B2 treatment, and B3 treatment were 14.53 ± 0.47%, 13.05 ± 0.13%, and 15.14 ± 0.07%, respectively. The cellulose content of B2 treatment was less than that of the other two treatments when the composting process ended (*p* < 0.05).

The hemicellulose contents in the three treatments first reduced and then increased (Figure 4B). The hemicellulose was degraded into monosaccharide by hydrolase in compost. The average hemicellulose contents in B1 treatment, B2 treatment, and B3 treatment were 13.93 ± 3.97%, 11.06 ± 2.61%, and 12.79 ± 4.69%, respectively. There were no significant differences in the contents of hemicellulose between the three treatments during the whole composting phase (*p* > 0.05). After 60 d of composting, the contents of hemicellulose in B1 treatment, B2 treatment, and B3 treatment were 10.80 ± 0.04%, 11.19 ± 0.03%, and 12.68 ± 0.04%, respectively. The hemicellulose content of B1 treatment was less than that of the other two treatments when the composting process ended (*p* < 0.05). B1 had effective degradation of hemicellulose in compost.

The changes in lignin in three treatments were almost the same, which were first increased and then decreased sharply (Figure 4C). The degradation of lignin was difficult in compost. The average lignin contents in B1 treatment, B2 treatment, and B3 treatment were 13.79 ± 1.67%, 14.54 ± 2.12%, and 13.63 ± 2.64%, respectively, and no significant differences were detected in the average lignin contents of the three treatments during the overall composting process (*p* > 0.05). When the composting ended, the lignin contents in B1 treatment, B2 treatment, and B3 treatment were 10.99 ± 0.03%, 14.11 ± 0.09%, and 11.20 ± 0.03%, respectively. The lignin content in B1 treatment was significantly lower than that in the other treatments when the composting ended (*p* < 0.05).

### 3.5. Changes in Bacterial Communities

The number of OTUs in B2 and B3 treatments gradually decreased during composting, whereas the number of OTUs in the B1 treatment reduced by day 29 and rose by day 60 of composting (Figure 5A). The average number of OTUs in the B1 treatment, B2 treatment, and B3 treatment was 897.33 ± 278.18, 848.33 ± 343.85, and 657.33 ± 297.52, respectively. There were no significant differences in the number of OTUs between the three treatments during the overall composting process (*p* > 0.05). Day 1 represented the thermophilic stage of composting, day 29 represented the cooling phase of composting, and day 60 represented the maturity phase of composting. When the compost entered the heating and thermophilic phase, the increasing temperature led to the reduction of the OTUs amount. B1 had the potential to enrich the number of bacteria in compost. Chao 1 index, Goods_coverage index, Observed_otus, Shannon index and Simpson index are shown in Appendix A, and sequencing data is shown in Appendix A.

The dominant bacteria were the same at the phyla level in the three treatments (Figure 5B), specifically, *Bacteroidetes* (39.10–76.82%), *Firmicutes* (13.56–51.17%), and *Proteobacteria* (3.03–25.90%) were the dominant bacteria in B1 treatment; *Bacteroidetes* (9.49–85.41%), *Firmicutes* (11.46–54.74%), and *Proteobacteria* (0.93–2.35%) were the dominant bacteria in B2 treatment; and *Bacteroidetes* (14.53–78.14%), *Firmicutes* (0.33–25.40%), and *Proteobacteria* (2.04–68.76%) were the dominant bacteria in B3 treatment. No significant differences in the abundance of *Bacteroidetes*, *Firmicutes*, and *Proteobacteria* was detected between the three treatments (*p* > 0.05).

The dominant bacteria are presented at the genus level in the three treatments in Figure 5C. *Muribaculaceae_unclassified* (10.24–33.00%), *Bacteroides* (5.53–18.40%), and *Escherichia-Shigella* (0.06–23.47%) were the dominant bacteria in B1 treatment; *Muribaculaceae_unclassified* (1.84–31.60%), *Bacteroides* (1.68–21.10%), and *Lachnospiraceae_NK4A136_group* (2.47–10.58%) were the dominant bacteria in B2 treatment; and *Bacteroides* (0.00–42.19%), *Fodinicurvataceae_unclassified* (0.00–40.07%), and *Muribaculaceae_unclassified* (0.00–13.40%) were the dominant bacteria in B3 treatment. The average abundance of *Lactobacillus* in B1 treatment, B2 treatment, and B3 treatment was 0.79%, 0.77%, and 0.53%, respectively, which indicated that inoculating the microbial agents at 1:1:1:1:2 could accelerate the reproduction of *Lactobacillus*, thereby enriching the abundance of *Lactobacillus* during the composting phase.

## 4. Discussion

*B. subtilis* could produce amylase and proteinase to accelerate the decomposition of organic matter, which is one of the most commonly used microbial agents in compost [30]. Moreover, *B. subtilis* has a strong ability to degrade the cellulose resulting from the produced cellulase [15]. *B. licheniformis* could shorten the time taken to reach maturity in aerobic composting, and it is helpful to effectively decompose the cellulosic substances in compost, thus *B. licheniformis* has a lot of potential to apply to compost [17,30]. A previous study stated that *T. viride* was inoculated to bagasse composting, meanwhile, *T. viride* could inhibit the production of the pathogenic bacteria in compost [31]. Incubation of *T. viride* has a role in improving the humification and compost quality [32]. *A. niger* has profound effects on enriching the compost quality and shortening the composting period [29]. Nakasaki et al. (2017) found that compost with added yeast played a pivotal role in degrading the organic matter [33].

This study reveals that the ratio of microbial agents at 1:1:1:1:2 can significantly increase the content of total organic carbon (*p* < 0.05). The results indicated that the microbial agents played a vital role in preserving carbon in B1 treatment. Mineralization and humification occurs during the whole composting process, mineralization is related to the carbon cycle during aerobic composting where organic substances are degraded by microorganisms [34]. The results find that the proportion of microbial agents at 1:1:1:1:2 increases the amount of OTUs and the relative abundance of *Lactobacillus* and the pH value was significantly reduced (*p* < 0.05) when the composting ended. It is noted that lactic acid bacteria secrete acid during composting [35,36]. pH is the important factor affecting the microbial activity and correlates with the carbon content during composting [37]. The increase in pH value may be due to the decomposition of nitrogenous compounds during the early composting period [38]. The reduction in pH value can be explained by these reasons: nitrification and volatilization [39,40].

The ratio of C/N, composting period, and maturity influence the mineralization [41]. The GI value significantly increases in B1 treatment (*p* < 0.05), indicating that the maturity of compost is improved, thus decreasing the mineralization. The humic substance carbon, humic acid carbon, and fulvic acid carbon are major fractions of carbon in compost [14]. The fact that the contents of humic substance carbon, humic acid carbon, and fulvic acid carbon fluctuated in the three treatments might result from the complex structure of the humus, leading to the uneasy occurrence of mineralization and polymerization [42]. Duan et al. (2020) found that inoculating *B. subtilis* at 0.5% significantly increased the total organic carbon and humic substances contents [14]. In our study, B1 treatment significantly enriches the content of humic acid carbon (*p* < 0.05), which reflects that the ratio of microbial agents at 1:1:1:1:2 improves the humification, thereby preserving the carbon in compost. The cellulose substances are degraded to other organic matter by microorganisms, which is beneficial to the formation of humic substance carbon [43], thereby promoting the conversion of humic substance carbon to humic acid carbon. Lignin is difficult to degrade in comparison with cellulose and hemicellulose during the composting process because the degradations are mainly implemented in the thermophilic composting stage by fungi and actinomyces, whereas the fungi and actinomyces die easily at such high temperatures [44,45]. B1 treatment significantly accelerates the degradation of hemicellulose and lignin (*p* < 0.05), thereby promoting the enrichment of humic acid carbon content. Jurado et al. (2015) reported that using inclusion of *B. licheniformis* and other bacteria as inoculants in compost facilitated the degradation of cellulose, hemicellulose, and lignin, and also promoted humification [46]. Similar results were observed in composting with inoculants (*B. subtilis* and *C. thermophilum*) [47]. In addition, the enrichment of total organic carbon may be due to the decomposition of cellulosic substances and the utilization of humic substances, which could be a mechanism to increase the carbon content in compost inoculated with microbial agents.

Carbon is utilized by microbes to maintain growth and metabolism during composting, and as a result between 34% and 77% is lost from the initial content of total carbon [48,49,50,51]. The results show that the content of total organic carbon increased instead of decreasing compared with the initial contents of TOC, and was enriched by 12.32%, 4.78%, and 1.90% in B1 treatment, B2 treatment, and B3 treatment, respectively. There are reports that the TCA cycle is a vital way to produce CO_2_ during aerobic composting, during which *Bacillus subtilis* impeded the synthesis of nucleotide purine and deamination, thus decreasing the release of CO_2_ into the compost [52,53]. The carbon conversion rate correlates with the emissions of gases, such as CO_2_, CH_4_, and other greenhouse gases [54,55]. Consequently, the low rate of carbon conversion results in the high contents of total organic carbon and humic acid carbon in B1 treatment.

## 5. Conclusions

In this study, *Bacillus subtilis*, *Bacillus licheniformis*, *Trichoderma viride*, *Aspergillus niger*, and yeast were inoculated in goat manure at ratios of 1:1:1:1:2 (B1), 2:2:1:1:2 (B2) and 3:3:1:1:2 (B3), respectively. The relationship between the TOC preservation and physicochemical indicators and bacterial communities were investigated during the composting process. The results found that the microbial inoculum with the 1:1:1:1:2 proportion significantly increased the total organic carbon in compost in comparison with the other two treatments (*p* < 0.05). The results of the physicochemical indicators indicated that the moisture content and pH value in B1 were significantly lower than those in B2 and B3 (*p* < 0.05), which may lead to the enrichment of carbon in B1 treatment. Furthermore, the results also revealed that B1 treatment could enrich the content of humic acid carbon (*p* < 0.05) and accelerate the decomposition of hemicellulose and lignin (*p* < 0.05). Another reason may be that the amount of OTU and the relative abundance of *Lactobacillus* were increased. These results are vital to the preservation of carbon in compost, and other mechanisms are still worthy of being investigated in further studies.

## Figures and Tables

**Figure 1 molecules-26-04441-f001:**
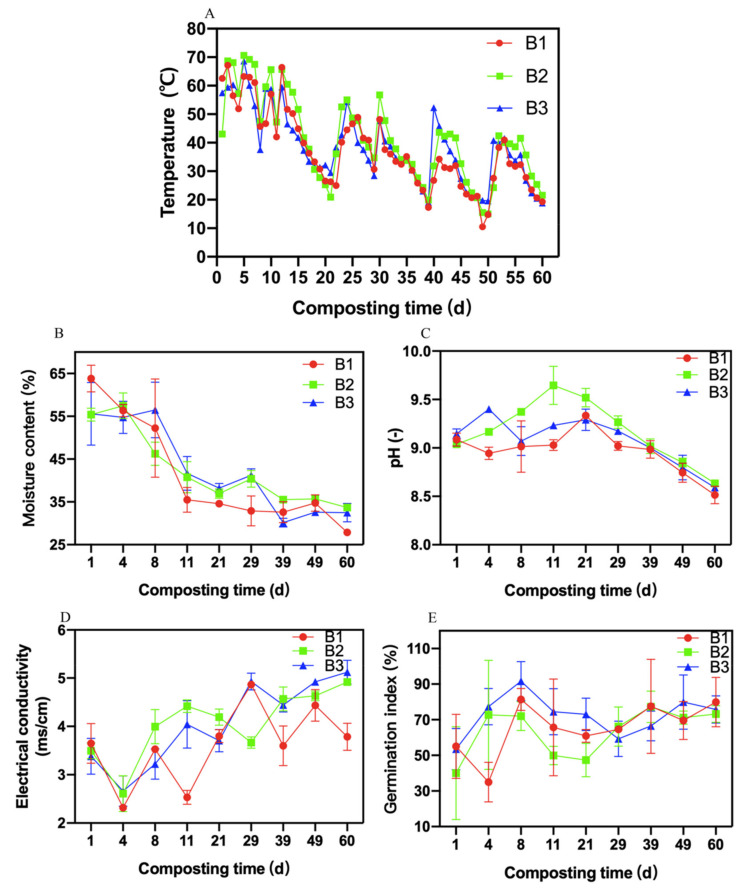
Changes in physicochemical indicators of compost under different treatments. (**A**): Temperature; (**B**): Moisture content; (**C**): pH; (**D**): Electrical conductivity (EC); (**E**): Germination index (GI). Error bars represent standard deviations of measurements (n = 3).

**Figure 2 molecules-26-04441-f002:**
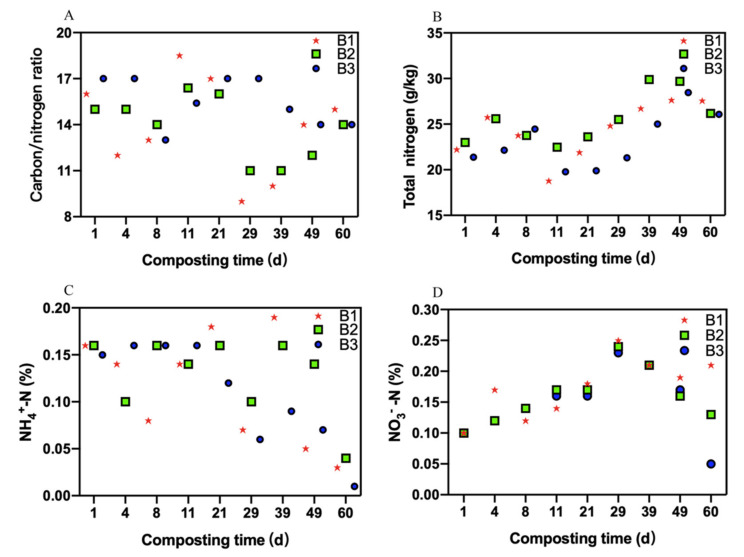
Variation in the contents of nitrogen components. (**A**): Variation in the carbon/nitrogen ratio (C/N); (**B**): Variation in the total nitrogen (TN) content; (**C**): Variation in the NH_4_
^+^-N percentage; (**D**): Variation in the NO_3_
^−^-N percentage.

**Figure 3 molecules-26-04441-f003:**
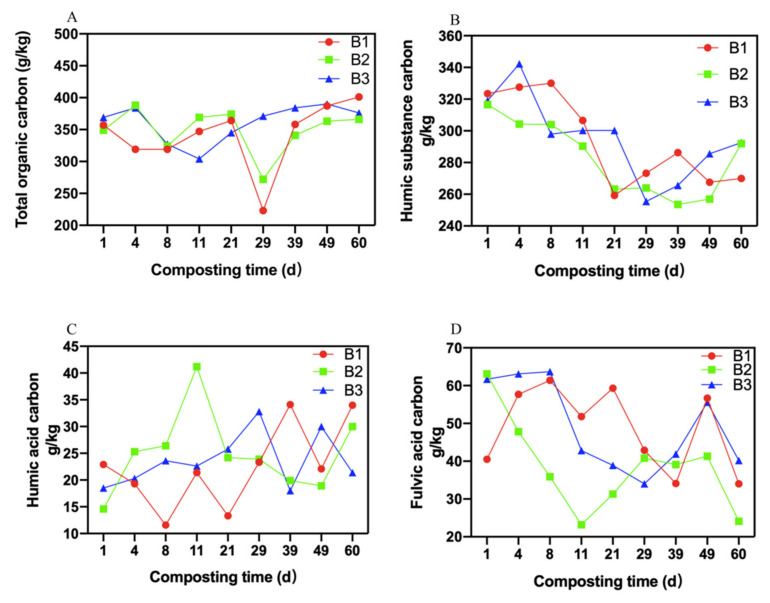
Variation in carbon. (**A**): Variation in TOC content; (**B**): Variation in HS-C content; (**C**): Variation in HA-C content; (**D**): Variation in FA-C content.

**Figure 4 molecules-26-04441-f004:**
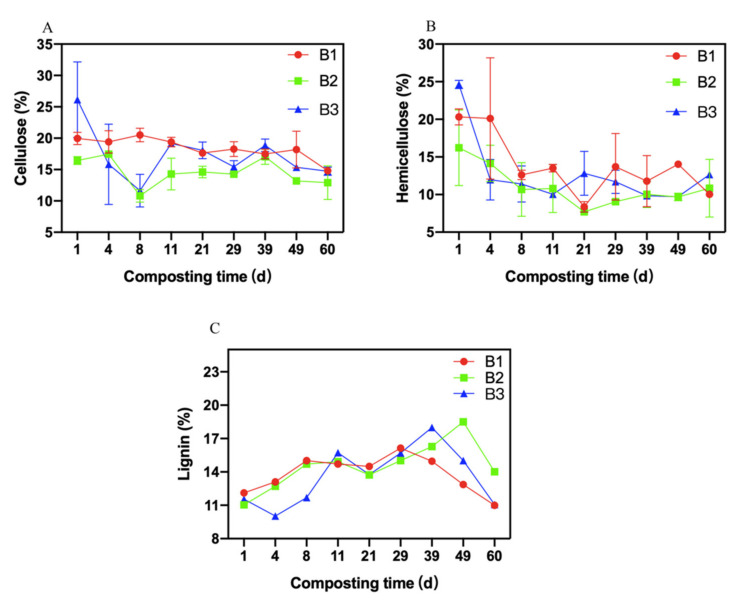
Variation in fiber. (**A**): Variation in the cellulose content; (**B**): Variation in the hemicellulose content; (**C**): Variation in the lignin content. Error bars represent standard deviations of measurements (n = 3).

**Figure 5 molecules-26-04441-f005:**
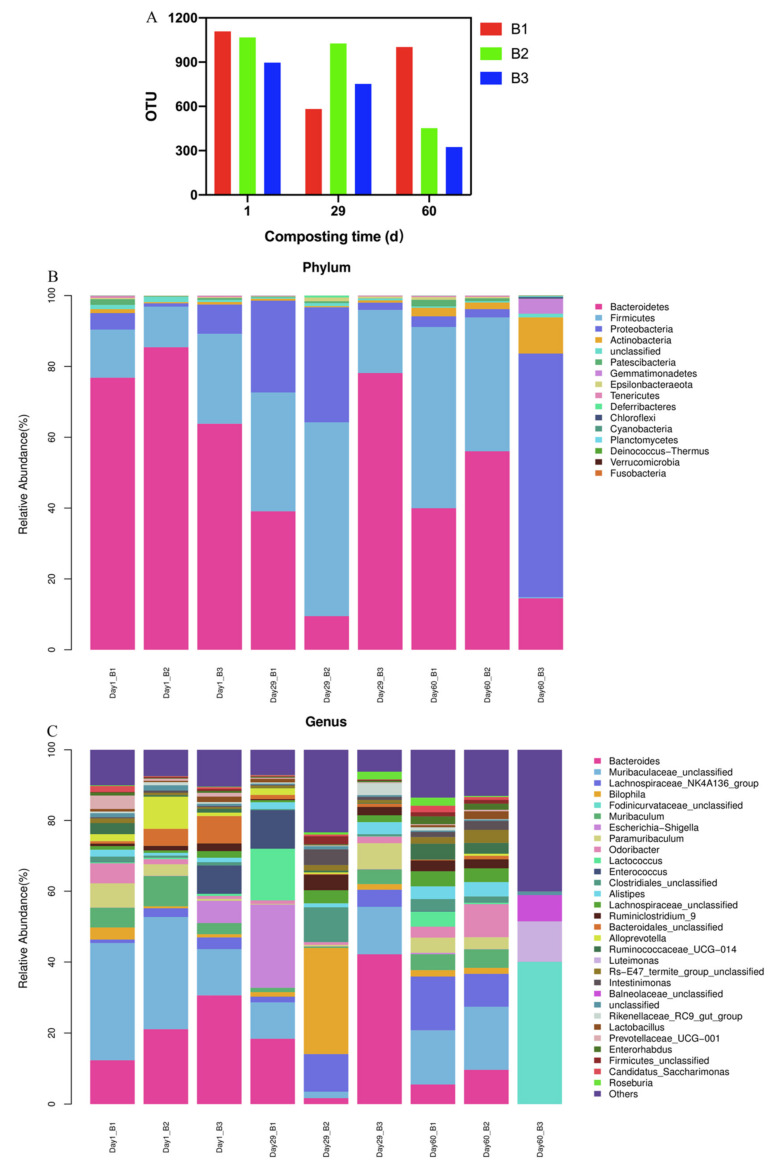
Variation in bacterial communities. (**A**): Variation in OTU amount; (**B**): Variation in the microbiota abundance at the phylum level during composting (Top 30); (**C**): Variation in the microbiota abundance at the genus level during composting (Top 30).

**Table 1 molecules-26-04441-t001:** Physical and chemical indicators of composting raw materials (moisture content, pH, total organic carbon, C/N, NH_4_
^+^-N, NO_3_
^−^-N).

Material	Moisture Content (%)	pH	TN (g/kg)	TOC (g/kg)	C/N	NO_3_ ^−^-N (mg/kg)	NH_4_ ^+^-N (mg/kg)
Goat manure	62.42 ± 4.35	8.26 ± 0.21	18.60 ± 2.31	276.00 ± 6.72	5.00 ± 0.49	199.00 ± 10.48	591.00 ± 13.44
Rape straw	9.59 ± 0.43	6.65 ± 0.17	4.86 ± 0.28	328.00 ± 3.52	67.00 ± 2.17	247.00 ± 12.31	301.00 ± 11.76

TN: total nitrogen; C/N: total carbon to total nitrogen; NO_3_^−^-N: nitrate nitrogen; NH_4_^+^-N: ammonium nitrogen.

## Data Availability

Not applicable.

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
