# Peer review of "Effect of Microbial Inoculation on Carbon Preservation during Goat Manure Aerobic Composting"

_molecules, 2021, doi:10.3390/molecules26154441_

Round 1

Reviewer 1 Report

In the paper, the author pointed effects of microbial inoculation on carbon preservation and the bacterial community during goat manure aerobic composting.
I accept a corrected manuscript. 
I recommend this article is worthy to publish in the journal. 

Author Response

We greatly appreciate the reviewer for your valuable comments concerning our manuscript entitled “Effect of microbial inoculation on carbon preservation during goat manure aerobic composting. The comments have greatly helped us in revising and improving our manuscript. We do hope our manuscript is now suitable for publication.

Reviewer 2 Report

In the manuscript “Effect of microbial inoculation on carbon preservation during goat manure aerobic composting”,  authors studied the carbon preservation inoculating microbial agents, composed of Bacillus subtilis, Bacillus licheniformis, Trichoderma viride, Aspergillus niger and yeast, during goat manure composting. The results provided a ratio of microbial agents to preserve the carbon during goat manure aerobic composting.

The structure and description in this manuscript was well prepared. The procedures, protocols, and methods used was well presented and described.  The manuscript is clearly written and structured. Morover the significance, the results and the impact of this study could be interesting.

I suggest minor revisions:

  • Delete the table 2, it is not necessary, the proportion of bacteria is clear in the text
  • Describe the methods of preparation of bacterial cultures (Times, temperatures, volumes)
  • -Why do you store the DNA at -80 ° C, it is not necessary ... it is not RNA!
  • Line 136 add “ipervariable” region v3-v4
  • Why do you use Qubit for DNA? it is not suitable for measuring purity and quality
  • The paragraph of Discussion is very short. I am recommended Needless information about the roles of Bacillus subtilis, Bacillus licheniformis, Trichoderma viride, Aspergillus niger and yeast, during goat manure composting.
  • The novelty of the work does not describe and more description needed.
  • In discussion the comparison of this work with other publication in this field does not performed.

Author Response

This manuscript is a resubmission of an earlier submission. The following is a list of the peer review reports and author responses from that submission.

Round 1

Reviewer 1 Report

One of the main objectives was to conserve nutrients. The objective was not achieved due to procedures,  presentation, calculations, and statistical analysis. This is not of the quality of information that this journal should accept unless the article is vastly improved. I believe that the goals of the article are worthy of investigation, but the outcome is poor or not completely presented. I would have the authors read one of their references carefully to see a better way to proceed with data they may have available to them. It is their #5 reference: Molecules, 2019, 24 2513.

Reviewer 2 Report

The manuscript " Efects of microbial inoculation on carbon preservation and the bacterial community during goat manure aerobic composting " is interesting for the scientific community but some aspects must be improved before considered for publication.

  1. The study lacks data on the methodology of sequencing and the methodology of the bioinformatic analysis (total number of reads, chimeric reads, etc.).
  2. How I prepared proportion of bacteria to fungi considering that these are completely different types of microorganisms. Why was yeast added that died at temperatures above 40 ° C?
  3. Why the presence of fungi and bacteria added to the compost was not monitored. The temperature in the compost was kept above 55 ° C, and the added fungi could not survive at this temperature. How the authors can prove that the bacteria and fungi added survived in the compost.
  4. How do the authors explain such large fluctuations in C: N (C: N = 9 on day 28; C / N = 19 on day 49?
  5. The raw sequence data should be deposited into the NCBI Sequence Read Archive.
  6. How was DNA isolated?
  7. Fig. 5 is too small, unreadable for the reviewer.
  8. The Materials and Methods section does not explain all the methods used in the research (eg GI). Chapter should be re-written.
  9. The entire manuscript should be corrected by a native speaker